# Tip Speed Ratio Optimization: More Energy Production with Reduced Rotor Speed

**Amir Hosseini [1], Daniel Trevor Cannon [2] and Ahmad Vasel-Be-Hagh [2,*]**

[1]  Independent Researcher, Mashhad 9189847849, Iran
[2]  Mechanical Engineering Department, Tennessee Tech University, Cookeville, TN 38501, USA
*  Correspondence: avaselbehagh@tntech.edu

**Abstract:** A wind turbine's tip speed ratio (TSR) is the linear speed of the blade's tip, normalized by the incoming wind speed. For a given blade profile, there is a TSR that maximizes the turbine's efficiency. The industry's current practice is to impose the same TSR that maximizes the efficiency of a single, isolated wind turbine on every turbine of a wind farm. This article proves that this strategy is wrong. The article demonstrates that in every wind direction, there is always a subset of turbines that needs to operate at non-efficient conditions to provide more energy to some of their downstream counterparts to boost the farm's overall production. The aerodynamic interactions between the turbines cause this. The authors employed the well-known Jensen wake model in concert with Particle Swarm Optimization to demonstrate the effectiveness of this strategy at Lillgrund, a wind farm in Sweden. The model's formulation and implementation were validated using large-eddy simulation results. The AEP of Lillgrund increased by approximately 4% by optimizing and actively controlling the TSR. This strategy also decreased the farm's overall TSR, defined as the average TSR of the turbines, by 8%, leading to several structural and environmental benefits. Note that both these values are farm-dependent and change from one farm to another; hence, this research serves as a proof of concept.

**Keywords:** wind farm; tip speed ratio; wake losses; optimization; renewable energy; wind energy

## 1. Introduction

### 1.1. The Wake Loss Problem and Its Significance

High-speed undisturbed wind enters front-row wind turbines of a wind farm. The wind turbine's large rotating blades extract the wind's kinetic energy and induce severe disturbances. Hence, wind leaving the turbine becomes a low-speed, highly turbulent plume called a "wake" [1]. Downstream turbines that receive an upstream wake at their inlet produce much less power than their upstream counterparts [2]Such "wake losses" can be as large as 70% in wind directions aligned with the column of turbines [3,4]. Wake loss is the most significant challenge the wind energy science faces and is the major cause of wind's low power density, defined as the amount of power generated per unit of Earth's surface area occupied by the power plant. A typical wind farm's power density ranges between 1 and 2 W/m², below natural gas (∼482 W/m²), nuclear (∼241 W/m²), oil (∼195 W/m²), coal (∼135 W/m²), solar (6–7 W/m²), and even geothermal (2–3 W/m²) [5]. This is one of the main reasons that wind energy amounts to only about 7% of the world's electricity generation after all the progress made in recent decades [6]. Expanding wind farms' power density and wind energy's contribution to global electricity generation requires viable solutions for reducing wake losses. This paper presents a potential solution to address this problem partially.

### 1.2. Existing Solutions: Wind Farm Layout Optimization and Active Control Strategies

The wake loss minimization attempts occur at both the design and operation phases [7]. At the design phase, Wind Farm Layout Optimization aims to place turbines within the

given perimeter in a way that minimizes the overall exposure of wind turbines to upstream wakes [8,9]. However, no matter how advanced and effective a wind farm layout optimization is, it is impossible to eliminate this exposure, and its consequent losses in a utility-scale wind farm with dozens of turbines and continuously changing wind directions [10]. Hence, the wind energy community is constantly investigating active-control strategies to weaken the upstream wake or steer it away from the downstream turbines to decrease the wake losses during the operation phase [11].

The most-researched strategy to actively optimize a wind farm operation is yaw control [12]. The strategy imposes an intentional yaw angle to all or some of the turbines of a wind farm. Doing so would decrease the energy production of the yawed turbines; however, it steers their wake away from specific downwind turbines, increasing their energy production. Research has shown that the gains outweigh the losses. A substantial majority of the literature focuses on how yaw control affects power production rather than annual energy production (AEP). Note that the percent increase in AEP and total power are not the same because calculating a wind farm's AEP depends on the frequency and duration that winds blow in each direction. These frequencies and durations are not distributed uniformly, and power production in some directions contributes more to the AEP. If those directions benefited less from the active control strategy, the benefit of AEP would be less than power. However, AEP is what really matters because the plant owners sell energy (not power) to the consumers. One study that considers the effect of yaw control on AEP is the field experiment done by Howland et al. [13]. They conducted a utility-scale field campaign to evaluate the effectiveness of the yaw control strategy and realized an approximately 0.3% increase in the annual energy production, serving as a proof of concept for the potential of yaw control to mitigate some of the wake losses. Another field measurement campaign investigated the impact of a yaw control strategy on a 9-turbine wind farm and found approximately 1% increase in AEP [14]. In addition to these field campaigns, a stochastic procedure estimated yaw control could lead to a 3% increase in a 9-turbine wind farm's AEP. The said procedure combined a generalized Polynomial Chaos technique and large-eddy simulations [15], while the literature finds yaw control a viable strategy to increase the energy production of wind farms by a few percent, some researchers have noted that yawing a rotor might induce additional loads on the turbine. For instance, Van Dijk et al. [16] found that yaw optimization increased blades' mean differential flap-wise and edge-wise moments by approximately 95% and 59%, respectively.

Other active strategies to improve a wind farm's performance by influencing the wake include tilt and pitch control. Tilt control can steer the wake away from specific downstream turbines [17]. For instance, Culter et al. [18] optimized the tilt angle across Princess Amalia 60-turbine wind farm, assuming that every turbine's tilt angle remained fixed for the lifetime of the farm. Then, they considered active tilt control. Optimizing fixed tilt angles resulted in a 2.77% increase in the AEP, while active tilt control resulted in a 13.64% increase. Pitch control, on the other hand, affects a wake's strength and can help the performance by weakening the wind speed deficit within the wake [19]. Researchers have also investigated some combinations of these strategies [20].

### 1.3. The Proposed Solution: TSR Optimization

***What is this article's proposed solution and its rationale?*** The proposed solution is to deviate from the TSR that maximizes an individual turbine's efficiency in the interest of the entire farm as a whole. Such deviation would decrease the adjusted turbine's production; however, it weakens its wake, increasing its downstream counterparts' output. Figure 1 presents a small sample of the data generated in this research to provide a better demonstration of the proposed solution. The figure illustrates the solution for a three-turbine section of a column within a wind farm for one wind direction aligned with the column. Figure 1a shows the case of maximizing every turbine's efficiency. The maximum achievable power coefficient for the studied turbines (SWT-2.3-93) is 0.4454, which one can achieve by setting

the TSR at 9.2. Turbine manufacturers provide such data. The first turbine produced 77.9 MWh per year in this wind direction. The overall amount of energy reaching the second turbine was 69.6 MWh, leading to its production of 69.6 MWh × 0.4454 = 31.0 MWh. The energy received by the third turbine throughout the year in this wind direction was 49.6 MWh, resulting in the production of 49.6 MWh × 0.4454 = 22.1 MWh. Hence, these three turbines' total annual energy production in this one direction was 131 MWh. Figure 1b shows the production of these turbines after adjusting the TSRs. The efficiency of all three turbines decreased to 0.3911, 0.3423, and 0.3638, respectively. Hence, the production of the first turbine decreased to 68.4 MWh. This increased the energy received by the next two turbines so that their production increased to 33.8 MWh and 32.9 MWh, although their efficiency had decreased. The total annual energy production in this wind direction increased to 135.1 MWh, 4.1 MWh (~3.13%) more than the baseline case.

Note that the decrease in the production of turbine #1 is not equal to the increase in the energy amount that turbine #2 receives. This is intuitive since, to calculate the energy that turbine #2 receives, one must plug the wind speed that this turbine experiences into the nonlinear power curve. The wind speed is a nonlinear function of the axial induction factor, which nonlinearly depends on the thrust coefficient. The thrust coefficient is also a nonlinear function of TSR. To calculate the amount of energy that turbine #1 loses, on the other hand, one needs to compute the turbine's power coefficient (efficiency) for the new TSR using their nonlinear Equation (curve) and apply that to the energy that turbine #1 receives. Hence, there is no reason for these two amounts (what turbine #1 losses and what turbine #2 gains), calculated through two different routes, to be equal or proportional. From a physical point of view, one can appreciate this by noting that the entrainment of the undisturbed wind into the wake to recover the deficit is a nonlinear function of the induction factor.

For any given wind farm, one must identify every turbine's optimal TSR that maximizes AEP for every wind direction upfront and form lookup tables to control the farm's TSR throughout the operation actively.

***What is the novelty and significance of this research?*** This study proves that optimizing each individual turbine's efficiency would not maximize the farm's AEP. The study demonstrates that a real-time optimization and control of TSR for every turbine and wind direction can save a significant amount of AEP while reducing the blades' rotational speed on average, which offers several environmental and structural improvements, including reduced noise, bird/bat accidents, and leading-edge erosion. In addition, this solution does not require any significant additional hardware upgrade and does not add to the load that blades experience.

***What is this article's approach to examining the effectiveness of TSR optimization?*** This research utilized the Jensen wake model and the Particle Swarm Optimization to find the optimal TSR of the turbines of a utility-scale wind farm for every wind direction with a 5-degree increment. The analysis shows a 4% increase in AEP and an 8% reduction in the farm's average TSR. These are both significant improvements. A detailed description of the investigated wind farm follows this section. Section 3 explains the employed methodologies. Furthermore, finally, detailed results are presented and discussed in Section 4.

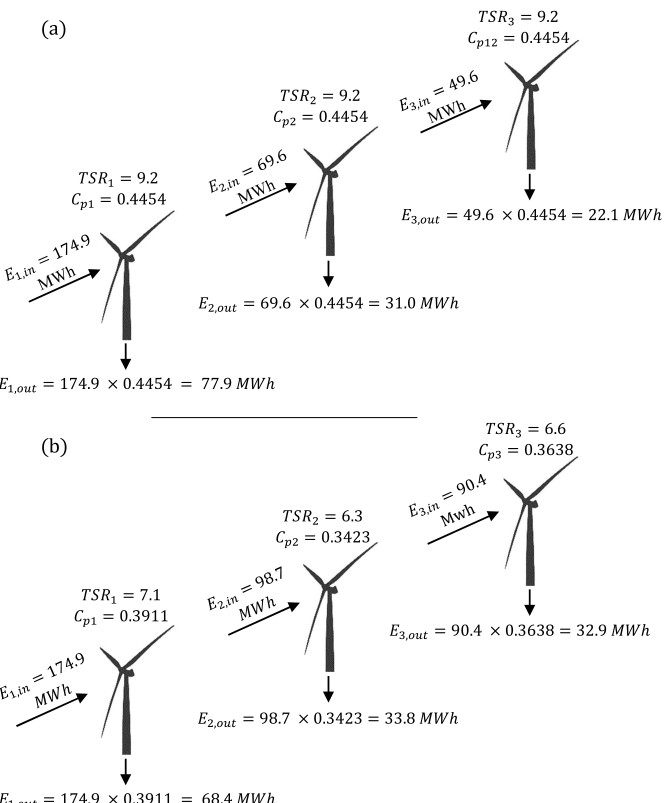

**Figure 1.** Solution: (**a**) optimizing TSR to maximize every individual turbine's efficiency, leading to a total energy production of 131 MWh, (**b**) optimizing TSR to maximize the total AEP, leading to 135.1 MWh.

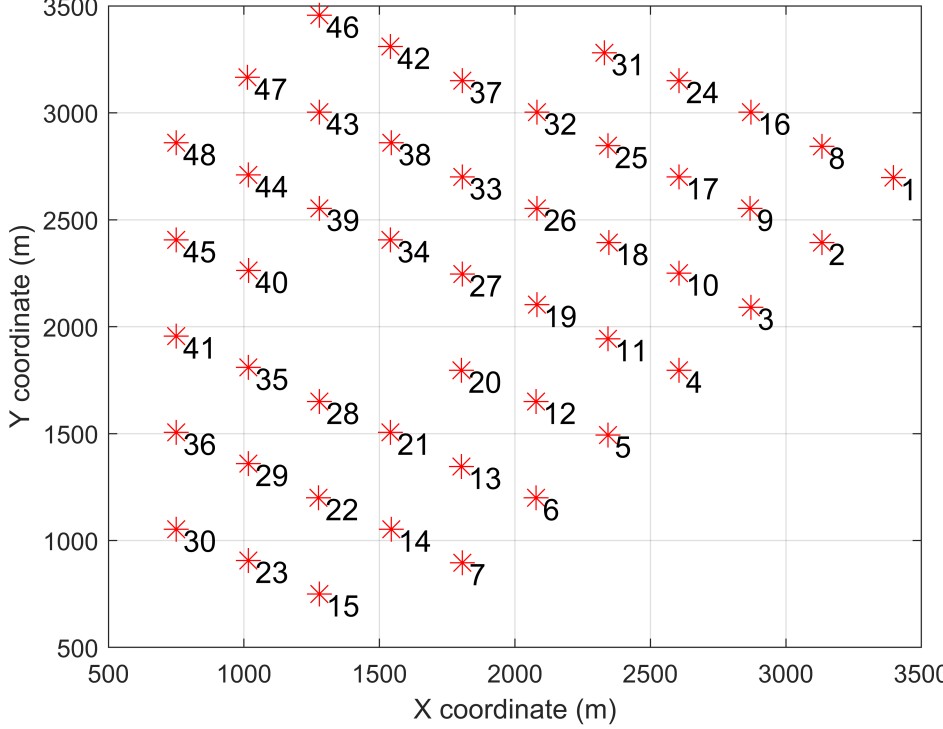

**Figure 2.** Lillgrund's layout.

## 2. Case Study

The study used Lillgrund, an offshore wind farm in Sweden, to investigate the solution proposed in Section 1.3. This wind farm includes 48 SWT-2.3-93 wind turbines, with a rated power of 2.3 MW and a rotor diameter of 93 m. To implement the proposed real-time TSR optimization, one needs to know the farm's layout, the power curve, the $C_t - TSR$ and $C_p - TSR$ curves, and the wind direction and speed distributions. Figure 2 shows Lillgrund's layout. Figure 3 illustrates the power curve of SWT-2.3-93 turbine used to calculate the energy production at every wind speed. The variations of power and thrust coefficients ($C_p$ and $C_t$) with TSR are given in Figure 4. Figure 5 presents the wind data collected at the hub height, i.e., 63 m. The wind speed distribution in every direction corresponds to Weibull parameters of $c_w = 9.42$ and $k_w = 2.41$.

## 3. Methodology

### 3.1. Optimizing TSR

Authors employed Particle Swarm Optimization (PSO), a bio-inspired algorithm with proven accuracy and high convergence speed for continuous optimization [21–25], to identify optimal TSRs for every turbine in every wind direction. The main idea behind the PSO method is moving a particle, which represents a candidate solution, around the search space to find the best possible solution. Employing a swarm of particles rather than one would accelerate the convergence.

In the present study, each particle represented an allowable solution in the form of an input vector ($V_{input}$) into the objective function $AEP = f(V_{input})$. This input vector was an $N_T$-long array of tip speed ratios, with $N_T$ being the total number of turbines, i.e., $V_{input} = [TSR_1\ TSR_2\ ...\ TSR_{48}]$. The goal was to find the global maximum of the objective function by moving the particles within the search space. The said movement of each particle was controlled by a three-term displacement vector shown as $\Delta V_{input}$ that defined the changes applied to every $\vec{V}_{input}$ vector at every iteration. This vector was computed as,

$$\Delta \vec{V}_{input} = \vec{V}_{inertia} + C_1 \vec{V}_{personal\ best} + C_2 \vec{V}_{global\ best} \tag{1}$$

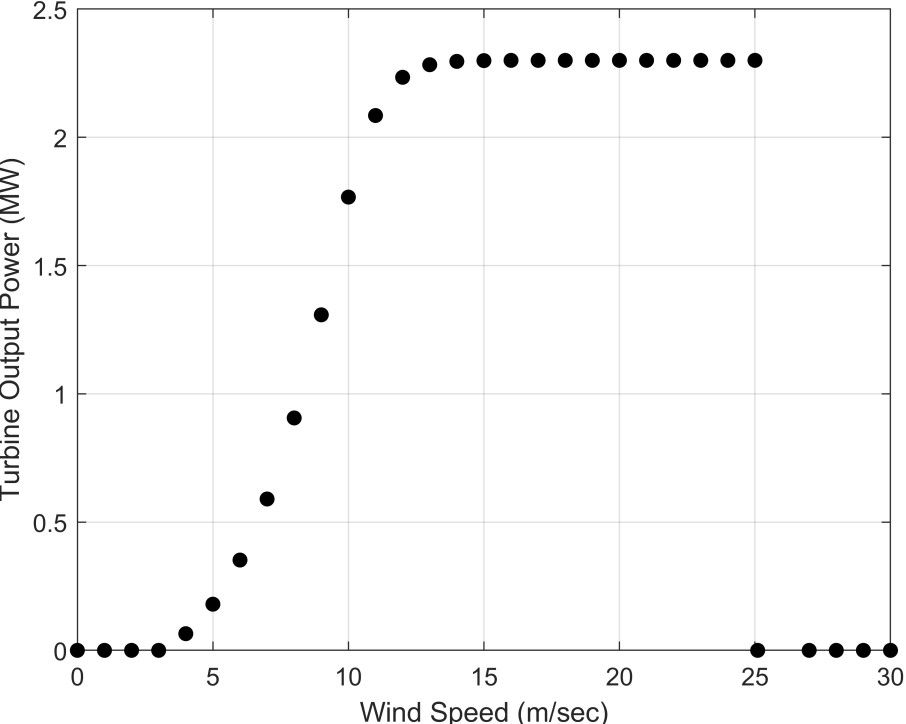

**Figure 3.** Power curve of SWT-2.3-93 turbine [2].

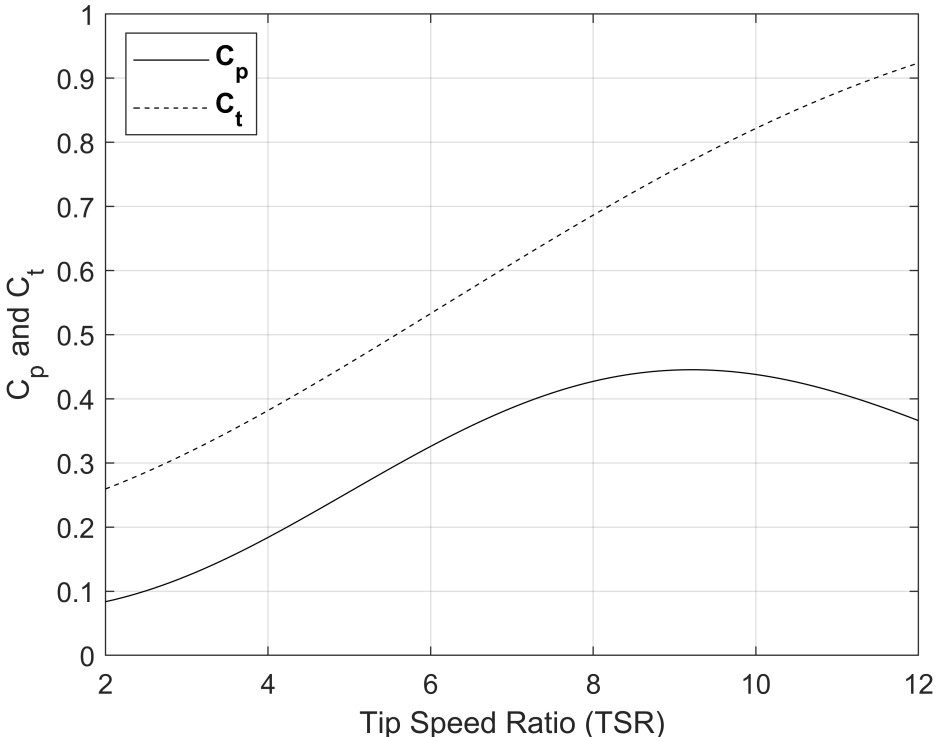

**Figure 4.** The $C_p$ and $C_t$ curves of SWT-2.3-93 turbine [26].

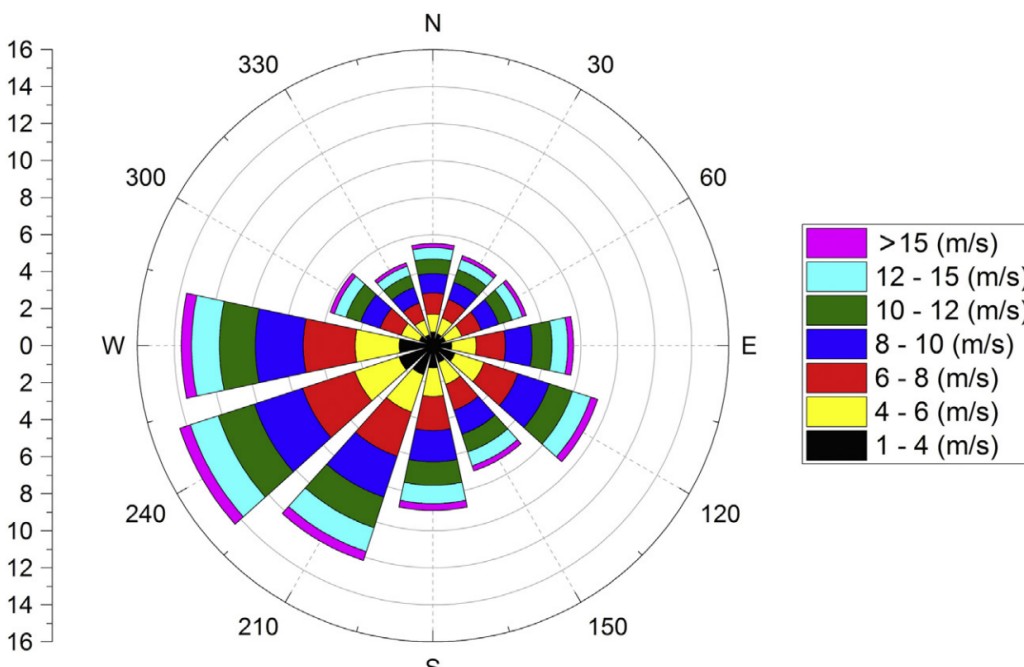

**Figure 5.** Wind data recorded at the hub height level ($h = 63$ m). Weibull parameters of the illustrated speed distribution are $c_w = 9.42$ and $k_w = 2.41$ [2].

This research defined $C_1$ and $C_2$, respectively, called cognitive and social coefficients, according to Maurice and Kennedy [27]. The inertia vector $\vec{V}_{inertia}$ had a random nature to ensure the particle's motion toward the optimum solution covered the entire domain of the solution. The personal best vector included the optimal solution found by the particle of interest, while the global vector introduced the best solution found by the swarm of

particles in the previous iteration. We updated $\vec{V}_{Personal\ Best}$ and $\vec{V}_{Global\ Best}$ in every iteration. After enough iteration with an adequate number of particles, the algorithm identified a TSR vector for every wind direction, resulting in a total AEP of 11 GWh (4%) larger than the baseline.

*3.2. The Jensen Model*

3.2.1. The Formulation

The authors employed the Jensen model to compute the impact of TSR on wake losses. This model is possibly the most widely used model [28] and performs reasonably well regardless of the wind turbine layout or wind direction [4]. Choosing an appropriate wake loss model is vital in predicting the power production of a wind farm and performing wind farm optimization. Due to their effective computational performance, analytical wake loss models are the most common candidates for such applications. A review of these models by Archer et al. [4] examined the performance of six well-known analytical WLMs, including Jensen, Larsen, Frandsen, Bastankah and Porté-Agel(BPA), Xie and Archer (XA), and Geometric Model (GM). This review compared each model's absolute error and bias of these models using observational data collected at three major commercial wind farms. Two of these wind farms are offshore (Lillgrund and Anholt), and one is inland (andNør-rekær ). One of these wind farms has a closely-spaced layout (Lillgrund), one employs a moderately-spaced layout (Nørrekær), and the third one is widely-spaced (Anholt). Two of these test wind farms have a structured layout (Lillgrund and Nørrekær), and the other is unstructured (Anholt). Hence, this evaluation considered wake loss models' performance over many conditions. Comparing all six models' predictions against observational data demonstrated that the Jensen and XA models stood out for their consistently strong performance. This was a major reason for using the Jensen model in this work. Cristina L. Archer, Ahmadreza Vasel-Be-Hagh, Chi Yan, Sicheng Wu, Yang Pan, Joseph F. Brodie, A. Eoghan Maguire, Review and evaluation of wake loss models for wind energy applications, Applied Energy, Volume 226, 2018, Pages 1187-1207.

Note that computational fluid dynamics cannot be used in the present study since this optimization would require so many runs, each of which would require approximately 2000 CPU hours.

Suppose one needs to compute the wind speed deficit experienced by turbine *i* caused by turbine *j*. This deficit is shown by $\delta_{ij}$. If the rotor diameter is *D* and the turbines are apart by an axial distance of $x_{ij}$, the Jensen model computes $\delta_{ij}$ as [29]:

$$\delta_{ij} = (1 - \sqrt{1 - C_t})(\frac{D}{D + 2k_w x_{ij}})^2 \qquad (2)$$

where $k_w = 0.04$ is the offshore expansion coefficient [30] and $C_t$ is turbine *j*'s thrust coefficient. According to the literature [4], one must correct the wind speed deficit as,

$$\delta' = (\frac{A_{overlap}}{A})\delta \qquad (3)$$

with *A* and $A_{overlap}$ being the rotor area and the fraction of the downstream rotor area covered by the wake from the upstream turbine. This corrected deficit needs to be computed for all upstream turbines that affect the turbine of interest. The inlet wind speed into the turbine of interest is then calculated as,

$$U_{in} = U_\infty[1 - (\Sigma_{j=1}^N \delta_{ij}'^2)^{\frac{1}{2}}] \qquad (4)$$

with N being the number of turbines upstream of turbine *i*. Knowing $U_{in}$ allows for calculating the turbine's power production using the power curve provided by the manufacturer (Figure 3).

### 3.2.2. The Validation

The authors conducted a validation study to ensure that the Jensen model's implementation was correct. The model was applied to a case of two NREL 5-MW wind turbines with an axial distance of 7D, with D = 126 m being the rotor's diameter, identical to that of Gebraad et al. [31]. Ten different cases were made by changing the lateral distance from −140 meters to +140 meters. Figure 6 compares the Jensen model's predictions of the power production of turbines 1 and 2 and the total power production with those obtained from large-eddy simulations. Data presented in this figure not only confirms the correct implementation of the Jensen model but also shows the reasonable accuracy of this simple model. This performance is not surprising since the literature heavily confirms the Jensen model's effectiveness in modeling a broad range of wind farms, from structured and unstructured layouts and packed to widely-spaced wind placements, both in aligned and non-aligned wind directions.

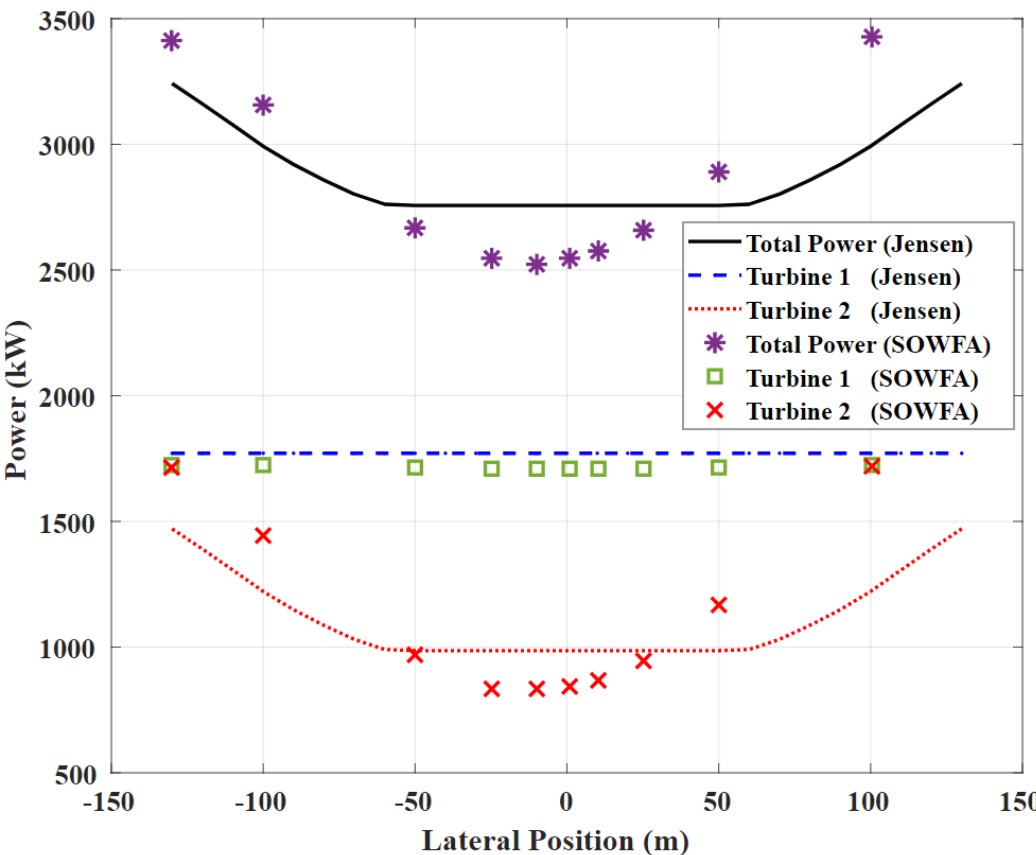

**Figure 6.** Comparing results obtained from our Jensen implementation and those of CFD [31].

### 3.3. Modeling the TSR Effect

Once the PSO algorithm described in Section 3.1 updated the input vector of 48 TSR elements in every iteration, the algorithm employed data presented in Figure 4 to calculate the $C_t$ and $C_p$ for every new TSR. It inserted the thrust coefficients $C_t$ into Equation (2) to update the wind speed deficit caused by each turbine. Then, using Equations (3) and (4), the new wind speed felt by each turbine was calculated and inserted into the power curve to compute the power produced by every turbine. This calculated power was then corrected by reducing it via $C_p/C_{p,max}$ factor. This correction is necessary since this turbine is no longer operating at its maximum efficiency because its TSR is adjusted. Power was then converted to AEP using wind direction and speed frequencies presented in Figure 5. The algorithm fed the results back into the PSO for it to update the input vector within the next iteration.

## 4. Results and Discussion

*4.1. Annual Energy Production*

It would be helpful to define the words "column" and "row" first since this discussion often uses them to refer to the location of turbines. A "column" refers to a group of turbines forming a line in the wind direction, while a "row" of turbines forms a line normal to the wind direction.

Consider the Lillgrund wind farm with $48 \times$ SWT-2.3-93 turbines with a $C_p - TSR$ curve provided in Figure 4. The industry's current practice is to keep the TSR of all turbines at 9.2 all the time and for every wind direction since it maximizes each turbine's power coefficient. The strategy presented in Section 1.3 claims that the idea of optimizing the efficiency of every turbine as a single, isolated unit does not necessarily maximize the total farm's power and energy production as a whole since this idea leaves out the dynamic interactions between turbines. This strategy suggests optimizing the TSR by accounting for such aerodynamic interactions can enhance the overall energy production. The wind farm TSR optimization presented in Section 3 was applied to Lillgrund and revealed that while lowering front-row turbines' TSR decreases their production, it grows the power production of their downstream counterparts. If done intelligently, the gain in the output of downwind turbines outweighs the upstream turbine's losses, leading to an overall production increase.

It is worth first focusing on one wind direction to put the proposed strategy into perspective. Figure 7 details the TSR optimization in the wind direction of 150° from the north using meteorological convention. In this direction, the algorithm suggested dropping the front-row turbines' TSR by approximately 21-23% (Figure 7a). It then decreased the TSR by an additional 9–15% from the front- to the second-row turbines and kept it approximately uniform until the last three rows. The algorithm raised the TSR within the last three rows so that the last-row turbines' TSR reached the baseline TSR of 9.2. Note that two of the columns partially deviated from this trend. The farthest left column had only three rows; hence, while, like for every other column, the algorithm dropped the front-row turbine's TSR by approximately 20% and assigned a TSR of 9.2 to the last-row turbine, the adjustment recommended to the second-row turbine's TSR was very slight (almost an additional 5%). The other unique column has a gap between its third- and fourth-row turbines. This gap was due to the shallow waters at that spot, preventing the developers from installing two of the Lillgrund turbines. That gap allowed for a better wake recovery; hence, the algorithm did not find it necessary to drop the other rows' TSR by too much. Like every other column, the TSR assigned to the last row was 9.2.

As expected, dropping the front-row turbines' TSR decreased their power and energy production. Overall, in this specific direction (150°), the summation of the power and energy drop in the eight front-row turbines was 0.75 MW and 62.47 MWh (Figure 7b,c). However, every other turbine's power and energy production increased. The power and energy gain became more significant as the wind moved towards the last row. The summation of all the power and energy gains appeared to be 5.34 MW and 439.78 MWh for 150°. Therefore, applying the wind farm TSR optimization in this wind direction led to a net gain of 4.55 MW and 377.31 MWh in power and energy production, equivalent to a 25.5% increase. Note that 150° is a direction of alignment (wind direction and turbine columns are aligned). Wake losses are generally much larger in the alignment directions; therefore, the strategies aimed to address wake losses appear more effective in such wind directions.

The authors applied the TSR optimization in every other wind direction using 5° increments. Figure 8 presents the impact of TSR optimization on the AEP for every wind direction. Given the above discussion on the 150-degree wind direction, the gains in that wind direction are highlighted to clarify how one needs to read and interpret these graphs. Summing up the net increase in AEP in every wind direction revealed that the TSR optimization increased the farm's AEP by 11 GWh, corresponding to about a 4% increase.

Furthermore, note that this study did not account for any delay in the turbine's response to the wind direction changes. The study employed a wind direction resolution

of 5 degrees to offset this to some extent. Optimizing the TSR using a higher resolution, such as 1 degree, would lead to more significant AEP gains; however, that becomes unreal since the turbines cannot respond to such fast wind direction changes in real-time. We assumed that the time it takes for the wind direction to change by 5 degrees is enough for the turbines to adjust their TSR to a new optimum. Furthermore, note that TSR control systems are generally fast-responding and can adjust to new wind directions and speeds pretty quickly [32].

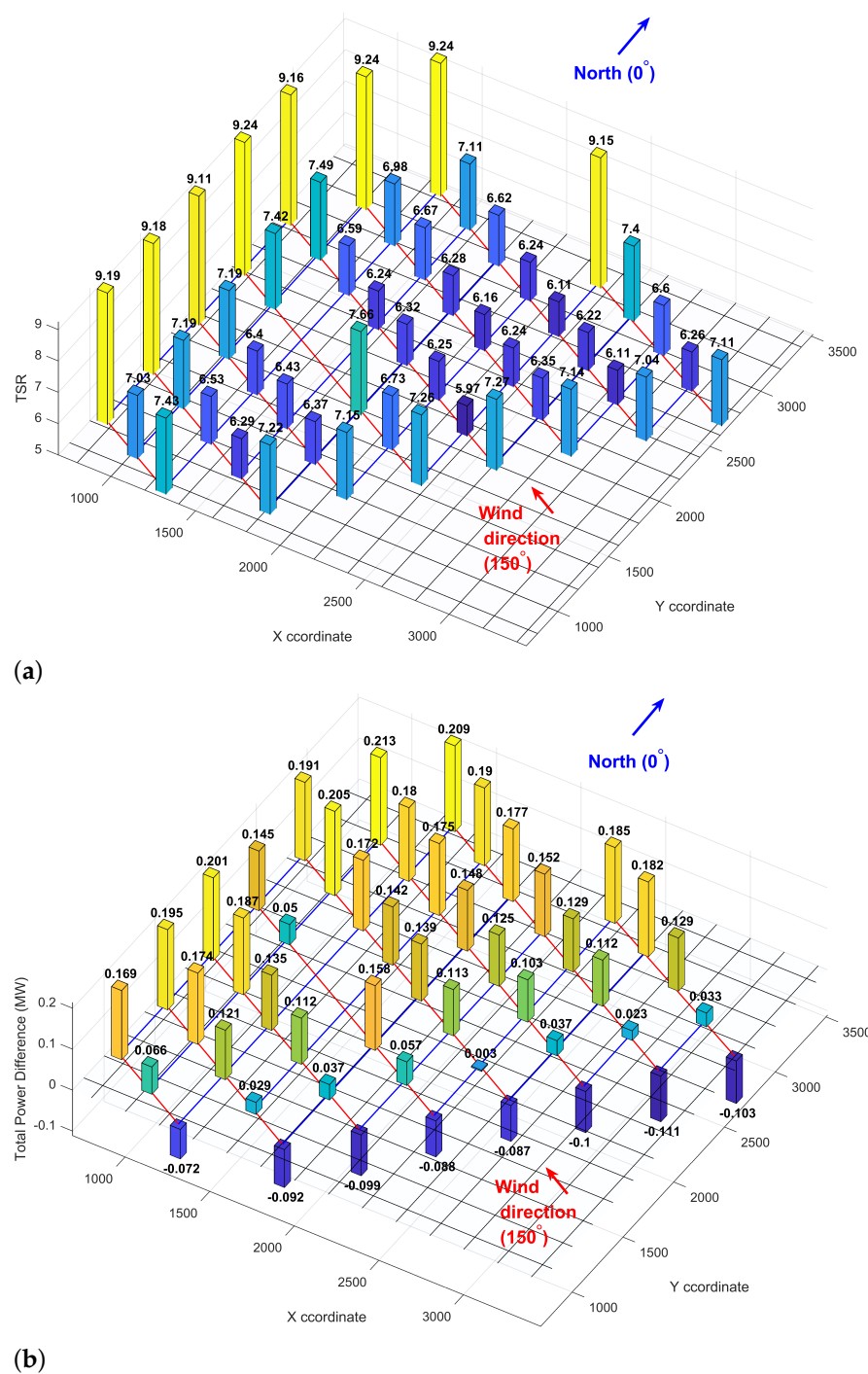

(**a**)

(**b**)

**Figure 7.** Cont.

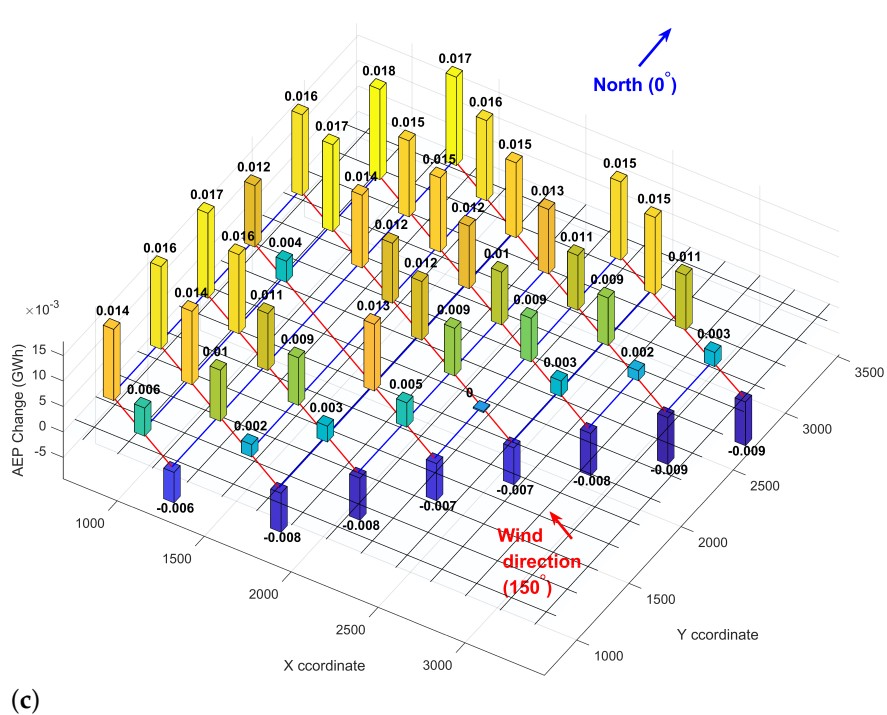

**(c)**

**Figure 7.** Optimizing TSR in 150°: (**a**) Optimal TSR values, (**b**) Changes in power production, (**c**) Changes in AEP.

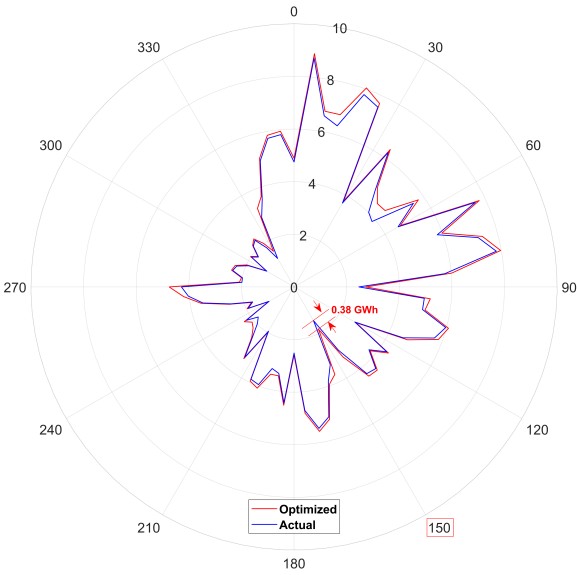

**Figure 8.** Wind farm's AEP (GWh) in every wind direction with and without TSR optimization.

It is essential to mention that there are several ways to control a turbine's TSR. Adjusting the pitch angle is one such way; however, that would alter the wake characteristics in other ways, including the wake direction. The present study aimed to understand the effectiveness of TSR active control alone and did not mean to test a hybrid TSR-pitch control strategy. There are alternative approaches to control the TSR that would not influence the turbine's wake. For instance, consider a direct-driven synchronous generator. In such generators, the voltage, passively rectified in a six-pole diode bridge, changes with the turbine's rotational speed [33]. Hence, one can control the turbine's rotational speed simply by controlling the DC voltage by either a DC/DC converter or by an active insulated-gate bipolar transistor rectifier [33]. Adjusting the external load applied to the generator is another option for controlling the TSR without needing any mechanical adjustments.

Appendix A provides optimal TSR of every turbine in every wind direction and its impact on power and energy production to share all the underlying data of Figure 8.

*4.2. Other Advantages*

Note that in addition to its significant impact on power and energy production, a real-time TSR optimization brings several other benefits, making it more exciting and promising.

- First, altering the TSR does not lead to any additional loading on the blades since the rotor still operates under normal conditions and is not misaligned in any direction. One only needs to increase the load applied to the generator to make it harder or easier to rotate. This can be achieved via electronics and does not require any mechanical modification.
- The proposed strategy appears to decrease the TSR overall. For the case of Lillgrund investigated in this article, the farm-averaged TSR decreased by more than 8%. A reduced TSR is equivalent to a slower rotor, and a slower rotor generates less noise since turbines' noise is primarily created by the giant blades cutting through the air, which is a serious environmental issue surrounding the wind energy industry [34,35]. So, an active TSR optimization not only enhances AEP, it reduces noise pollution.
- Slowing down the rotor helps reduce bird and bat collisions, which is a serious issue that needs to be addressed. Recently, an energy company was given five-year probation and ordered to pay approximately $8 million in fines as their wind turbines caused the death of 150 bald and golden eagles [36]. Decreasing a wind farm's overall TSR can help with such accidents.
- The proposed strategy enhances the performance of wind turbines by relaxing the leading-edge erosion (LEE) phenomenon. LEE is the deterioration of a wind turbine blade's leading edge by airborne particles such as sand, dust, rain, and insects [37]. Such erosion decreases the blade's lifespan and aerodynamic efficiency, which eventually reduces the farm's AEP. Slowing down the blades via TSR optimization contributes to addressing such LEE-induced issues.

## 5. Conclusions

Wind turbines adversely affect each other via their aerodynamic wake, an expanding, highly-turbulent, low-speed region behind them created by the rotating blades. The wake's significantly adverse effect on the wind farm's energy production is holding the wind power industry back from contributing more to the world's electricity generation. This paper proposed and investigated the real-time optimization and active control of TSR as a viable solution to reduce the wake effects. The paper showed that optimizing every individual turbine's efficiency would not maximize the farm's energy production. Instead, a subset of turbines must operate at a lower efficiency to allow turbines downwind to produce more power and boost the farm's production. Applying this strategy to an offshore wind farm with 48 × 2.3 MW turbines increased the annual energy production by approximately 4%. Note that it is not too challenging to actively control the TSR since all the hardware required to execute such a plan is already available at every wind farm. Furthermore, note that TSR control does not increase loading on the rotor and decreases the farm's TSR overall by 8%. Reduced TSR helps with environmental concerns such as noise production and bird/bat collisions. It also helps with the leading-edge erosion phenomena.

**Author Contributions:** Conceptualization, A.H., D.T.C. and A.V.-B.-H.; methodology, A.H. and A.V.-B.-H.; software, A.H.; validation, A.H.; formal analysis, A.H. and A.V.-B.-H.; investigation, A.H. and A.V.-B.-H.; resources, A.H.; data curation, A.H. and A.V.-B.-H.; writing—original draft preparation, D.T.C. and A.V.-B.-H.; writing—review and editing, D.T.C., A.H. and A.V.-B.-H.; visualization, A.H. and A.V.-B.-H.; supervision, A.V.-B.-H.; project administration, A.V.-B.-H.. All authors have read and agreed to the published version of the manuscript.

**Funding:** This research received no external funding.

**Institutional Review Board Statement:** Not applicable.

**Informed Consent Statement:** Not applicable.

**Data Availability Statement:** The data that support the findings of this study are available from the corresponding author at avaselbehagh@tntech.edu upon reasonable request.

**Acknowledgments:** Not available.

**Conflicts of Interest:** The authors declare no conflict of interest.

## Appendix A. Detailed power and energy production data

The data provided in this appendix covers the impact of the proposed TSR optimization on the power and energy production of each turbine for every wind direction. Figure A1 shows the power production of each wind turbine before and after optimizing its TSR within the wind farm. Figure A2 reflects each turbine's annual energy production in every wind direction. Figure A3 presents each turbine's optimal TSR in every wind direction that leads to the power and energy changes provided in Figures A1 and A2.

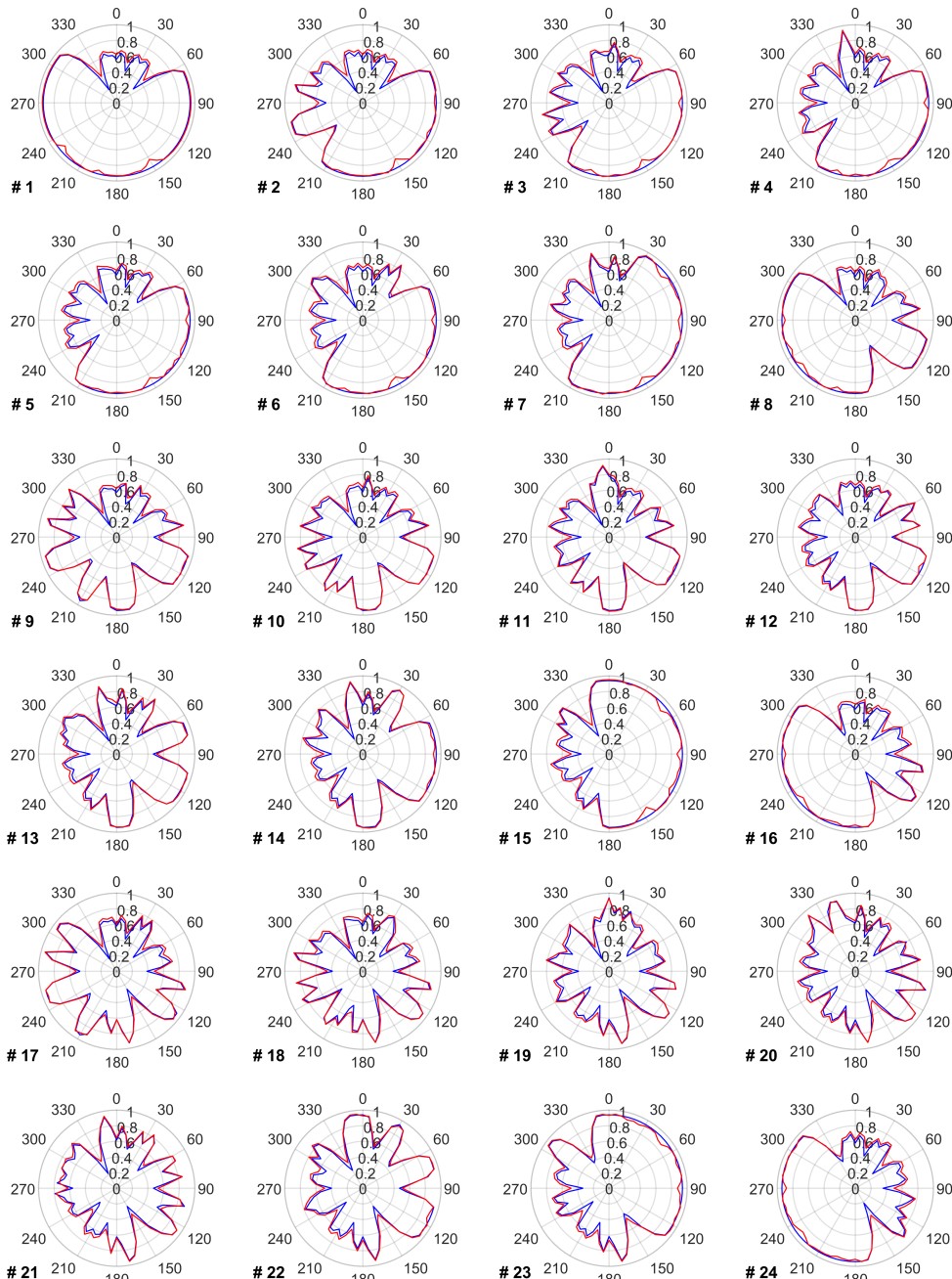

**Figure A1.** *Cont.*

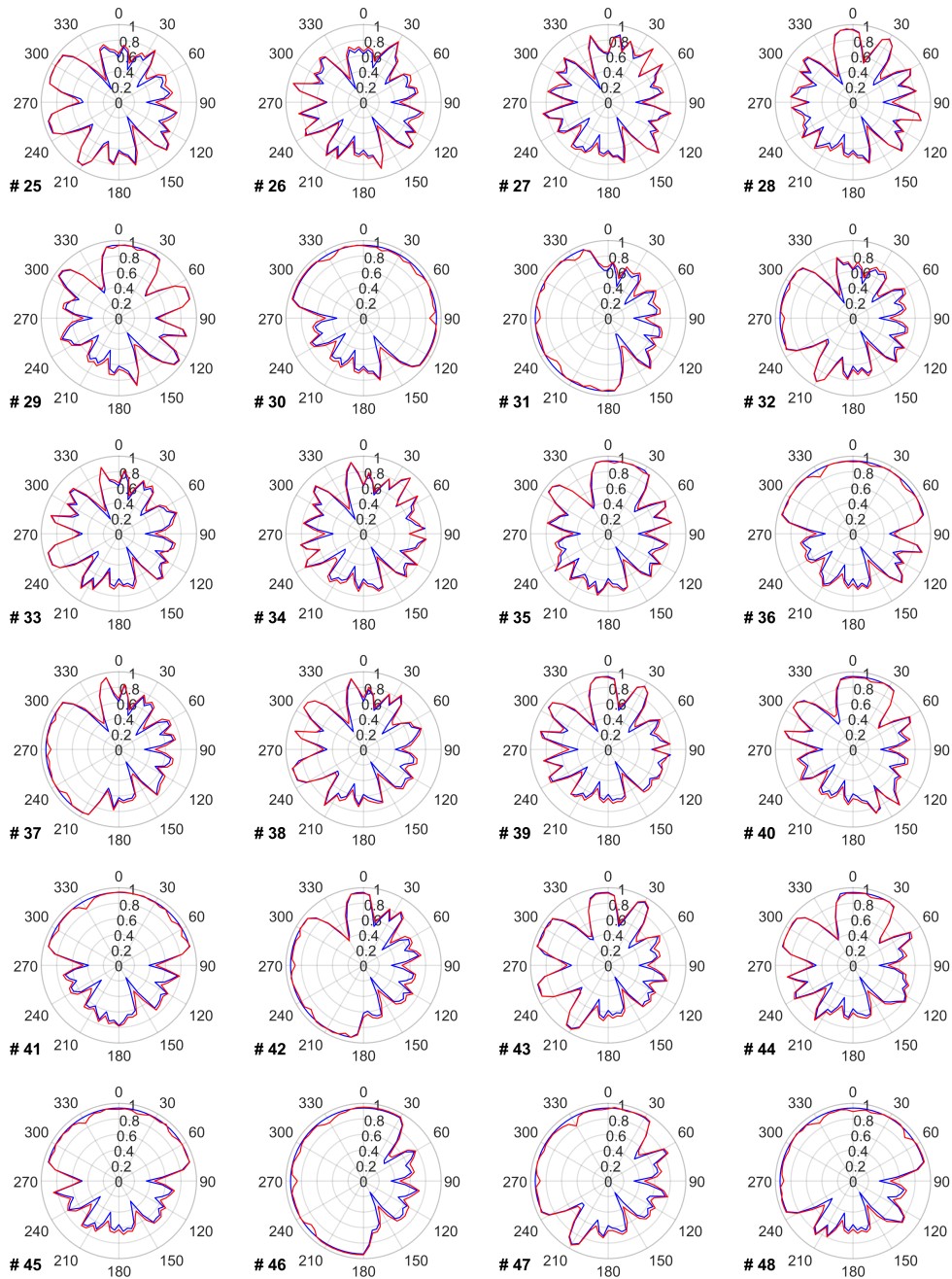

**Figure A1.** Blue line shows each turbine's relative power production in every wind direction without applying a TSR optimization. Red line shows the relative power production after applying a TSR optimization.

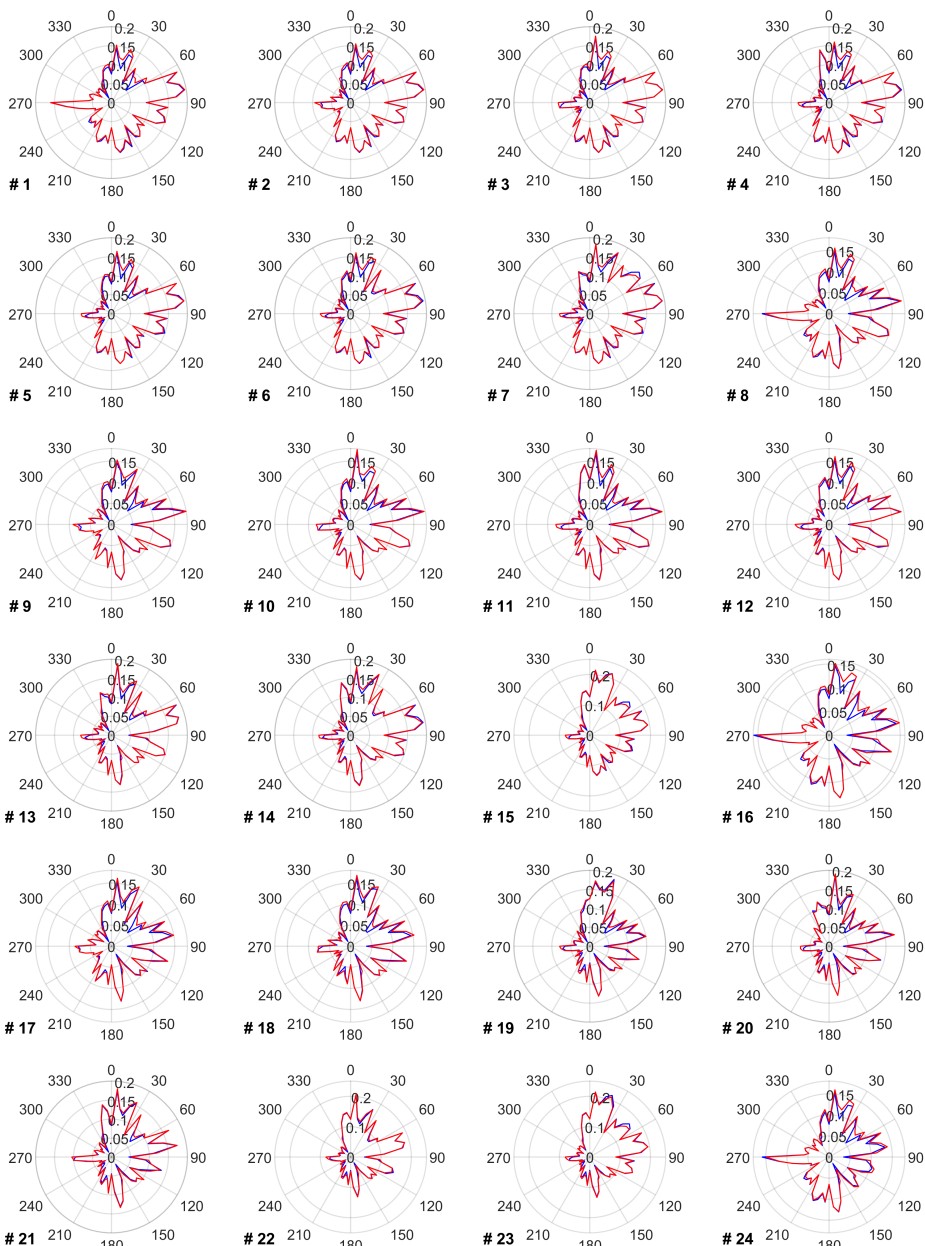

**Figure A2.** *Cont.*

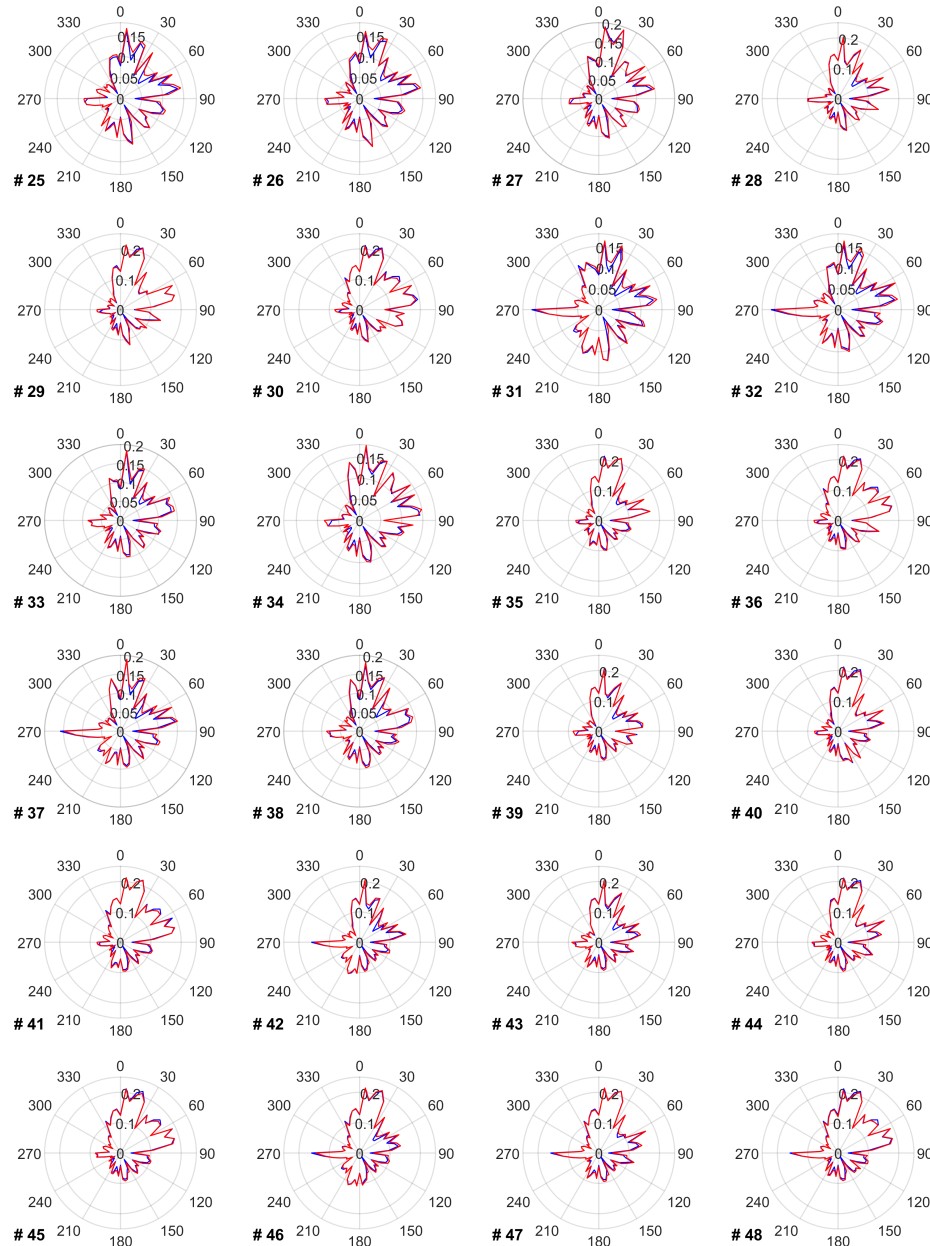

**Figure A2.** Blue line shows each turbine's AEP in every wind direction in GWh without applying a TSR optimization. Red line shows the AEP after applying a TSR optimization.

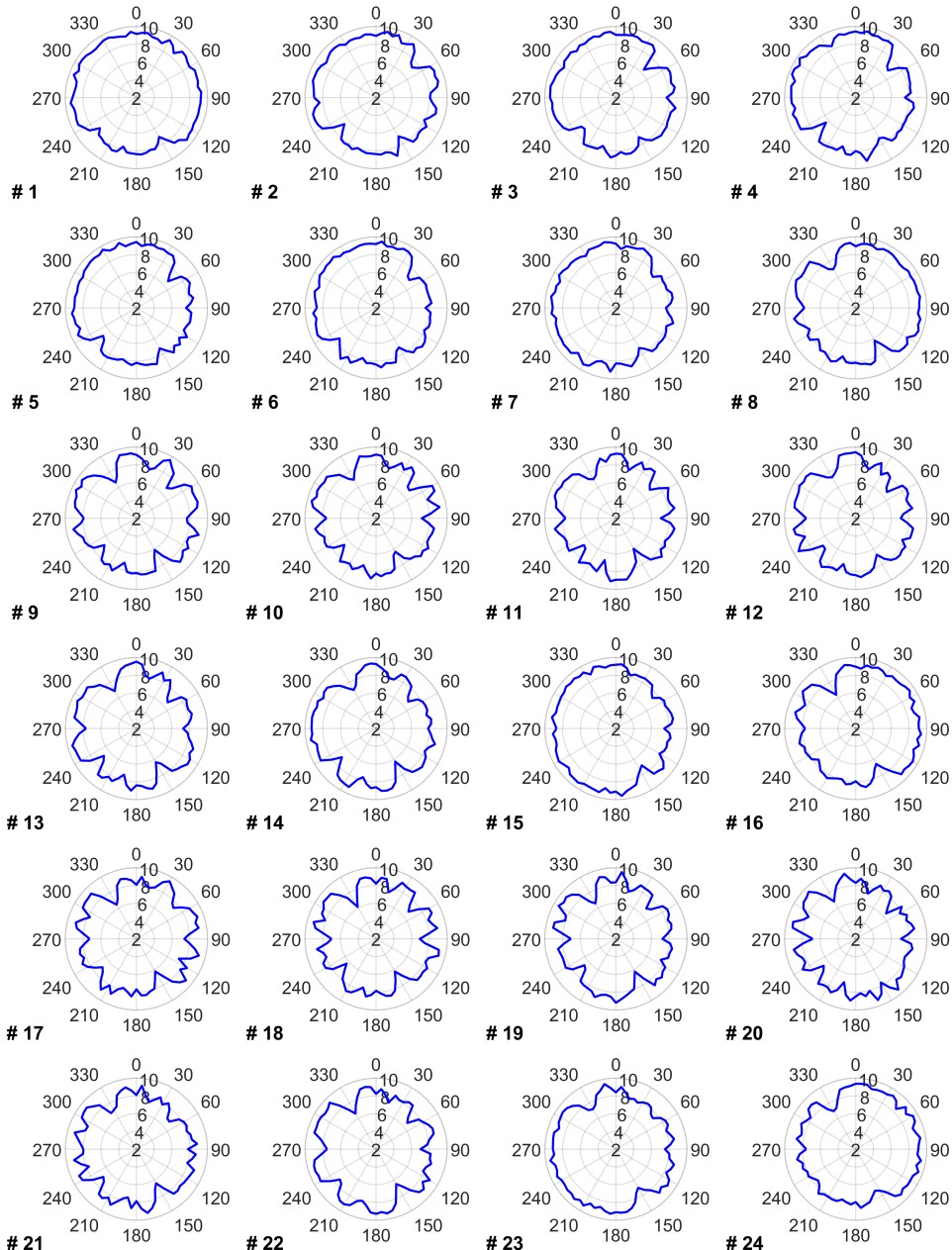

**Figure A3.** *Cont.*

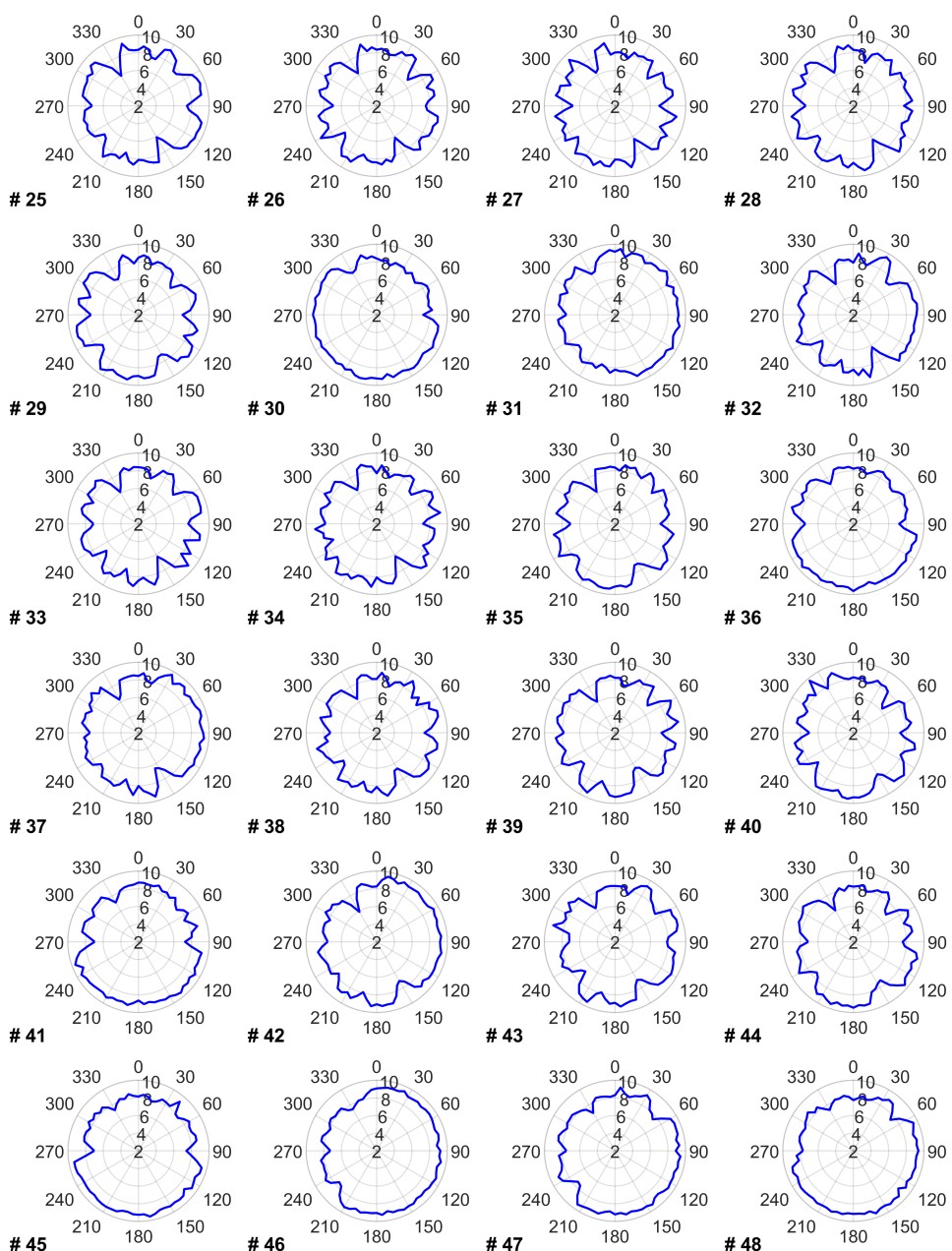

**Figure A3.** Optimal values of TSR for each turbine in every wind direction.

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
