# Peer review of "Tip Speed Ratio Optimization: More Energy Production with Reduced Rotor Speed"

_2674-032X, doi:10.3390/wind2040036_

Round 1

Reviewer 1 Report

I have included my comments in the attached file

Reviewer 2 Report

The article presents high quality in terms of both structure and content. The reviewer would like to positively assess the language, the motivation presented and a very well thought and organized flow of the scientific investigation. Minor comments are presented below:

L3: “The industry’s current understanding is that maximizing the efficiency of every individual turbine of a wind farm would maximize the annual energy production of the farm as a whole. Hence, they impose the same TSR that maximizes the efficiency of a single, isolated wind turbine on every turbine of wind farm.” – This might be a little bit too radical statement, wind farms already utilize control algorithms for efficiency maximization, You also presents these techniques in the introduction

L14: farm’s overall TSR – can You be more precise? Is it an average TSR from all units?

L34: …global electricity generation 33 requires viable solutions for reducing wake losses. – it’s difficult to consider the proposed solution as “wake loss reduction”, please reformulate

General question regarding the layout of the article – shouldn’t Figures 1 appear after being referred to in the manuscript? (Fig.1, Fig.2 appears before being mentioned)

Fig.1 – There is no MWh unit next to Ein values

L95 – in some formulas You omit MWh, please unify

L102 – please give the percentage improvement as well (4/131=3%)

L132 – used -> investigated

Figure 5 – did the authors create this figure?

L202 – fantastic is a slight exaggeration, differences reach 7% - 8%

After responding to above mentioned doubts, the article seems to be ready for publishing.

Round 2

Reviewer 1 Report

I advise publishing this paper because the authors satisfactorily addressed the questions.